# Identifying Limits in Domestic Law Delivering Net Ecological Benefit: A New Zealand Example

## Stephen Knight-Lenihan

Independent Researcher, 21773 Malmo, Sweden; s.knight-lenihan@outlook.co.nz

**Abstract:** Achieving a real net ecological benefit requires among other things legislative changes to existing environmental laws. New Zealand is one country undertaking such a review. The proposed new laws recognise a need to enhance the quality of the environment as a move away from minimising harm. As such, this appears to be a move toward a Positive Development (PD) approach to environmental management. However, as this paper concludes, the shift remains incomplete partly because while science is used to inform the creation of policies, plans, legislation and regulation, this is only achieved up to a point. That point is where the socio-economic norms and expectations prevent the on-going application of what is required by science to address observable and quantifiable ecological degradation. The understanding and application of ecological integrity is used as an example of how this can result in legislation apparently enabling significant change and a possible net ecological benefit but failing in effect to do so. The article concludes that legislative changes can better frame the problem of on-going ecological decline within the dominant paradigm, and as a result, it may deliver benefits, but these will not be net benefits in the Positive Development sense.

**Keywords:** biophysical capacity; nature positive; net positive development; environmental limits

## 1. Introduction

Storytelling can be used to highlight certain truths. Aldo Leopold in *A Sand County Almanac* observed that a thing is right when it tends to preserve the integrity, stability and beauty of the biotic community. However, it is wrong when it tends to do otherwise. This makes sense partly because it sits within the context of the story that is told in the book [1].

Presenting the facts about how much of the world's resources are being over-used to achieve human well-being (e.g., see [2]) is far less effective than taking people back to a forest where the bird and frog song they knew in their youth has now all but disappeared. This may trigger solastalgia, or the ecological grief associated with environmental degradation [3]. However, successive generations assume that what they were born into is normal, and the loss and degradation itself may be normalised (see, for example, the discussion by Pitcher [4] in the marine context).

Exemplifying this, when the Europeans arrived in 1770 in what is now New Zealand's Marlborough Sounds, the dawn chorus was "certainly the most melodious wild musick (sic) I have ever heard" [5] (p. 23). Now, the decimated choir in the surviving forests is sotto voce, and this is considered to be normal, with most of the change having occurred over two centuries [6].

Even though we know this to be true, undesirable, and in opposition to integrity, stability and beauty, it does not prevent us from supporting systems that contribute to continuing ecological decline. This is because there are two opposing systems with a differing internal logic; one of them concerns economics, growth and human welfare, the other one concerns ecological processes, biological welfare and ecosystem functioning and capacity. For example, the Intergovernmental Panel on Climate Change recently noted that "the very processes that have contributed to current climate challenges, including economic growth and the resource use and energy regimes it relies upon, are also the pathways to

improvements in human well-being. This places climate resilience and development in opposition to one another" [7] (p. 2671). Similarly, Otero et al. [8] observe that there is increasing evidence that economic growth contributes to biodiversity loss, but the majority of international biodiversity and sustainability policies still advocate for economic growth.

Therefore, we are dealing with two realities; one in which we know the Earth's biological systems could be collapsing, and the other one in which we need to earn money, get the kids to school and deal with the economic and political reality. Mental and social survival means a tendency to focus on the second reality and ignore the first one. The clash is well recognised, as exemplified by the recent parody of this clash in the 2021 film *Don't Look Up*.

A symptom of this dissonance is that the scientific analysis of planetary collapse may be accepted politically, economically and socially up until the point where the second reality intrudes. For example, the online tool Climate Action Tracker compares climate change mitigation targets, policies and actions against the Paris Agreement to hold global warming well below two degrees centigrade, and preferably, 1.5 degrees. CAT points out the gap between what is require by science to meet the Paris Agreement and the current commitments and action, which are a function of trading-off between what is identified as necessary by science and what is politically acceptable.

This applies to ecological decline, generally. A scientific analysis may be used to inform the creation of policies, plans, legislation and regulation up until the information cannot be absorbed by the existing socio-economic system. At least, that is the contention in this article.

To examine this idea and how it relates to Positive Development (PD; [9]), the example is used of New Zealand's current (as of 2022) review of the legislation governing planning and environmental management. The review incorporates a recognition of the need to enhance the quality of the environment and achieve positive outcomes [10] as a move away from minimising harm. As such, New Zealand's legislative review is relevant internationally as a case study of legislation that is being drafted which appears to adopt elements of PD.

The article explores the extent to which the political implications of the scientific analysis are followed through, the extent to which the legislation is likely to generate genuinely improved ecological outcomes, and the point where the scientific analysis meets socio-economic reality.

The phenomenon of there being two realities is a specific challenge to PD and its goal of compensating for accumulating ecological losses and the 'overshoot' of biophysical limits by increasing ecological space. In this context, ecological space provides for the maintenance, support and improvement of biodiversity values and ecosystem functioning, including allowing systems to change and evolve, such that human development over-compensates for current and accumulating losses and becomes nature positive [9]. The PD goal requires a policy to be formed according to scientific logic, the first reality, while the legislation is formed within a system that is moderated by the second reality of the socio-economic system. Using terminology that is consistent with Birkeland [9], the first reality is referred to as the PD frame, while the second one is the Dominant Paradigm (DP) frame.

## 2. Method

There are three questions addressed in this article:

Is the proposed legislative reform in New Zealand likely to deliver on the objectives stated in the legislation (the within-DP logic frame)?

Are the stated objectives anyway sufficient to deliver real net ecological benefit (the crossover from the within-DP to within-PD frames)?

What might the legislation look like when it takes a Positive Development approach (the within-PD frame)?

The first part of the paper summarises the legislative review processes, and it updates a 2021 paper by this author [11]. Components of the proposed legislation falling within the DP frame are described, answering the first two questions above. The draft legislation is then reviewed using Birkeland's method for interrogating sustainable design to assess the legislation's potential to deliver eco-positive outcomes [9]. Finally, a more PD-compatible version of the proposed reformed legislation is presented.

A comparison is made to the United Kingdom's *Environment Act 2021* which provides for English planning permission being granted on condition that the development contributes to improving biodiversity values. The UK is a neoliberal democracy that actively participates in the global economy [12], conditions which also apply to New Zealand.

Internationally, it is expected that other legislation and overarching policies will be promoting an ecological benefit. Therefore, a full review of such initiatives would be useful. However, the objective in this paper is to examine, in detail, a legislative review from a PD perspective which presents lessons for other countries undertaking similar reviews. Therefore, a full comparative review is not undertaken.

The review is not urban specific, but it is directly relevant to urban systems, which generate significant resource demand and waste impacts far outside of their geographical area.

## 3. The Draft Legislation

Currently, the main planning legislation in New Zealand is the *Resource Management Act 1991* (RMA) which was introduced to promote the sustainable management of resources while protecting the life support capacity of the natural and physical environment and accounting for the future generations (s5).

The New Zealand Government is currently replacing the RMA with a Natural and Built Environment Act (NBA), a Spatial Planning Act (SPA) and a Climate Adaptation Act (CAA). In November 2022, a Natural and Built Environment Bill (NBEB) and a Spatial Planning Bill (SPB) were tabled in the New Zealand Parliament. The CAA bill was not available at the time of writing. The NBEB and SPB are available for public feedback and political debate, which will continue into 2023, before the final legislation is passed.

The genesis of the draft NBA was a government-established Resource Management Review Panel recommending, inter alia, that the NBA should focus "on enhancing the quality of the environment and on achieving positive outcomes to support the wellbeing of present and future generations" [10] (p. 5).

The Environment Minister David Parker says the RMA created barriers to development while failing to protect the environment due to it having " . . . too narrow a focus on managing the negative effects of resource use rather than providing direction on desired environmental outcomes" [13] (p. 3). In contrast, the NBA "aims to improve on the RMA through setting up a framework of outcomes for restoring, enhancing or improving the natural environment where it is degraded. It will also promote development outcomes within environmental limits . . . " [13] (p. 3). There will be a greater emphasis on understanding environmental interconnectedness to better manage cumulative effects and a recognition of the "essential relationship between the ecological integrity of the natural environment and its capacity to sustain all life and the economy" [13] (p. 4).

The purpose of the NBEB (s3) is to enable the use, development and protection of the environment in a way that supports the well-being of present and future generations. This is to be performed at the same time as managing adverse effects, promoting environmentally beneficial outcomes and complying with environmental limits and targets. In addition, the act would recognise and uphold the interconnectedness of all of the parts of the environment, the essential relationship between the health of the natural environment and its capacity to sustain all forms of life and the intrinsic relationship between the Maori people and the environment. The mitigation hierarchy of avoiding causing harm and reducing the effects will be maintained, as it exists in the RMA, while more explicitly allowing for the offsetting or compensation (termed redressing, s14) of the residual environmental impacts.

A National Planning Framework (NPF) and local council plans (s5) must provide for, among other things, the protection, or if it has been degraded, the restoration of the ecological integrity of the air, water, soils, coastal environments, wetlands, estuaries, lakes and rivers and their margins and indigenous biodiversity. The NBEB defines ecological integrity as the ability of the natural environment to support and maintain the occurrence and extent of ecosystems and indigenous species and their habitats, the natural diversity and abundance of indigenous species, habitats, and communities, the biotic and abiotic physical features of ecosystems, and the ecological and physical functions and processes of ecosystems (s7).

The NPF sets environmental limits, targets and "strategic directions" (NBEB s33(c)), including those that are integrated with environmental management (33(a)). The targets can also be set directly using local plans (s48). The targets for unacceptably degraded environments will be set through the NPF (s50), and communities can elect targets that go beyond this.

Meanwhile, the purpose of the Spatial Planning Bill includes promoting the integration of the NBA with the legislation and regulations covering infrastructure and long-term planning (SPB s3(b)). Coupled with the proposed Climate Adaptation Act, the objective is to identify where development, growth and infrastructure should be, the areas needing protection and the areas affected by or likely to be affected by climate change.

While the proposed legislation enables the protection and enhancement of the environment, the mechanism put forward of establishing environmental limits and targets, is not the same as one requiring an improvement to its ecological values. Limits and targets tend to suit static systems rather than dynamic ones, thus reflecting a narrow view of what is meant by biophysical systems. In terms of the dominant paradigm, this will not necessarily generate expected improvements in the natural environment.

In the proposed legislation, the environmental limits must relate to ecological integrity or human health, and they must be set as the minimum biophysical states or the maximum amount of harm or stress to the natural environment (NBEB s40). Combined with the NBEB's purpose, which was noted earlier, this sets up conditions for the trade-off between human wellbeing and ecological integrity, despite the earlier definition of ecological integrity encompassing the conditions that would seemingly be required to ensure human health. This seems to be a contradictory position.

The limits are set to prevent ecological integrity degrading from current conditions (s37), and they are to be set in relation to the conditions of the air, biodiversity, various habitats, and soils. Presumably, it is acceptable for the systems to accumulate until they reach these impact capacity limits or bottom lines. Where this becomes unacceptable is when aspects of the natural environment are degraded and require restoration (s5). This is where the limits have been exceeded, and these aspects of the natural environment will need to be brought back to within those limits.

While superficially providing legislative support for creating net ecological benefits, its success hinges on what is meant by degraded and what baselines are used. Ongoing monitoring shows that most of the aspects of the natural environment have been degraded and are continuing to do so, with some localised exceptions of successful protection and restoration [14]. Given that trading-off will occur, arguably, the limits and targets will reflect what is seen as normal in a socio-economic context, rather than what might be deemed necessary to address the accumulated ecological loss.

There is also a need to take an integrated approach across the environmental domains listed in s37. Minister Parker said that "sufficient consideration" [13] (p. 5) will be given to integrated management, as well as cumulative effects and the precautionary principle. This is reflected in the wording of the NBEB, which notes that decision makers must provide for the integrated management of the environment. However, it is unclear how this is to be achieved.

Using water as an example, successfully improving freshwater quality and ecological functioning requires improving land use practice. This has consequent impacts on

coastal waters. It is necessary to consider the interaction between terrestrial ecosystem functioning (of which biodiversity is a measure), soil condition (which in turn benefits from and contributes to the ecosystem's functions), and consequently, freshwater, wetland, coastal and marine health (of which, again, biodiversity is a measure). Such integrated management has property rights/existing use rights implications. A consideration of this level of integration and its implications is not explicitly identified as being required in the legislation.

This is a particular example of how a logical process identifying a way to address the need to halt and reverse ecological decline strikes difficulties when coming up against the logic of the dominant paradigm. The broad integration necessary to achieve good ecosystem outcomes cuts across existing assumptions about property ownership and administrative boundaries, which significantly slow progress toward addressing ecosystem decline. This clash between ecosystem processes and human boundaries and institutions has long been debated in the literature (see, for example, [15] for a New Zealand perspective).

It is possible that the NPF and proposed Spatial Planning Act, due to the need to provide for the integrated management of the environment, will generate more debate regarding the extent to which existing use and property rights may need to be reviewed in order to meet the defined limits. This hinges on where baselines are set, and what are considered to be acceptable levels of such things as indigenous biodiversity and ecosystem functioning.

It should be noted that under the RMA, integrated land use and water management are already evolving through a national policy statement on freshwater management, and this and other RMA national direction instruments will be carried over into the new legislation (NBEB Schedule 1). However, these efforts in practice, as measured by the monitoring outcomes [14], are not as successful as it was hoped that they would be. This is discussed further below under Section 8: A Systemic Problem. A lack of space precludes there being a detailed discussion here, but a contributing factor to this is the clash between existing use property rights and the need to manage ecosystems coherently.

In conclusion, as the definition of ecological integrity in the NBEB includes the ability of the natural environment to support and maintain biotic and abiotic features and functions, and given existing levels of degradation as reported by the New Zealand Government agencies [14], this would seem to be what should drive the proposed legislation. Instead, this description of what underpins the continuation of life on the planet is relegated to one of a number of issues that must be taken into account. Consequently, ecological integrity is the focus of this paper's analysis of the existing proposed legislation and what it might look like under a PD frame.

## 4. Incorporating Ecological Integrity

Treating ecological integrity as one of a number of factors which must be taken into account within limits and baselines dramatically over-simplifies what ecological integrity means. The NBEB's wording suggests it is something that can be measured and achieved. What the legislation does not make clear is that its measurement and achievement rely on agreeing on what integrity is, and in addition, that it can be quantified and managed, which are two things that are highly contentious.

Integrity is a complex evolving interaction of sub-systems that defies a simple process of identifying relevant limits or baselines [16]. A measure of this complexity is the lack of global consensus on how to assess ecological integrity and its fundamental relationship to thermodynamics and self-organisation [17].

Including such a term in legislation assumes that some measurable assessment can be applied in practice, when such an assumption is, currently, misleading at best. At worst, it implies that ecological integrity can be controlled and delivered using available metrics, and so it can fit within a rational planning discourse and remain a tradeable item. To paraphrase the IPCC in the context of development and emissions [7], this fails to

recognise that improving integrity conflicts with business-as-usual decision-making, given trading-off to date has resulted in continuing ecological decline.

The difficulty for those drafting the legislation is how to include such an important concept while leaving what achieving it looks like open to interpretation. Perhaps oddly, referencing the likes of Leopold ("a thing is right when it tends to preserve the integrity . . . ") may provide a clue. Integrity is not an end-point, but an ongoing process, which is why the NBEB, as it is currently drafted, is inadequate.

If the environmental limits relate to either or both ecological integrity and human health, and if integrity is dynamic and multi-dimensional across the environmental "domains" or disciplines arranged to make the information manageable, rather than accurate portrayals of the reality, the limits will not necessarily result in improving the ecological functions. The challenge is to set dependable and predictable limits, allowing the business of consent granting to continue, while accepting that the actual ecological integrity outcomes will be unpredictable but positive.

The reason that this is so problematic within the DP frame is that under a benefit–cost approach, there is a need to measure the return on investment, while accounting for the costs of doing business. This applies to investing in environmental compensation [12], where what is fair and reasonable is like a yardstick: you want to compensate for what you effect, but no more. There is a need to develop planning tools which enable the measurement of the benefits to human welfare as part of increasing integrity: you invest in ecological assets that keep getting better, and you may obtain significantly better results relative to your impact. One approach is incorporating the idea of increasing ecological space [9], as discussed in the following section.

## 5. Critiquing Legislation from Positive Development Principles

Realising Positive Development would result in an increase in the ecological space and mechanisms to assess the whole-of-system impacts. This would include accounting for the upstream and downstream impacts associated with resource extraction, storage, fabrication, manufacturing, construction, maintenance and operation and disposal. In addition, it would use development in, for example, urban areas to increase the opportunities for the ecosystem processes.

Critically, PD theory argues that development should increase the ecological space to over-compensate for the existing and accumulated impacts, and these should become nature positive [9]. Normally, negotiations for a consent aim to be proportional to the impact of a proposed development. This means avoiding, and then, mitigating the ecological harm, with the possibility of compensating for any more-than-minor residual effects (the mitigation hierarchy). This approach is retained in the NBEB.

Following on from these two points, there are two fundamental and related problems with the NBEB. The first one is that within its own terms of reference, the proposed act sets up an expectation of enhancement, but it fails to put in place the mechanisms for ensuring this will occur (as discussed in the previous two sections). The second problem exists at a deeper level. The proposed act is inadequate in terms of recognizing that human activity needs to not just work within the biophysical limits, but also to extend the limits and restore processes. The provisions in both the NBEB and the Spatial Planning Bill allow for such outcomes to occur, but they are not required, and they are presented as part of a trading-off process. Extending the role of the limits so that they contribute to increasing the temporal and spatial extent of the ecosystem's functions requires fundamental changes in how human activity is managed. This requires a shift away from seeing the legislation's primary role as contributing to managing the natural and physical environment as part of ensuring human well-being towards seeing the legislation's primary role as ensuring ecological well-being by managing the human activity in a way that contributes to improving the whole system.

Continuing to set limits is a practical response to dealing with rapidly evolving whole-system problems, as described by Birkeland ([9] (p. 11)): "Since complex systems cannot be

measured, 'system boundaries' and limits are necessary within current forms of decision making (comparing and making choices), accounting (adding costs, risks, and benefits), and law (defining duties and expectations). These generally exclude problem solving and opportunity creating by design".

As a first step toward creating legislation that moves beyond the system boundaries, Table 1 compares selected PD standards from [9] (column 1) with the new system which is to be created by the draft Natural and Built Environment and the Spatial Planning acts and the proposed Climate Adaptation Act (the NBA system). The standards were selected on the basis that they are governance-related and potentially addressable through legislation. This excludes the standards relating more exclusively to design.

**Table 1.** Positive Development standards enabled, potentially addressed or not addressed by the proposed Natural and Built Environment Act, the Spatial Planning Act, and the Climate Adaptation Act (the NBA system). Section descriptions in column one are from Birkeland [9].

| Positive Development Standard | Included in NBA System | Comment |
|---|---|---|
| **Democratic Standards (s 7.2.2)** | | |
| Create direct universal access to natural systems and eco-services that provide means of survival, enable self-reliance, and prevent military, government or market monopolies on supplies to deprive citizens of genuine political or basic life choices. | Potentially and partially. | Creation of environmental limits and targets contributes to establishing ecological integrity, and they are based on minimum biophysical states. No reference to enabling self-reliance or ability to ensure that there are basic life choices. |
| Ensure public education and transparency about existing decision frameworks so that there is full public awareness of tacit anti-ecological biases in many decisions concerning environmental issues, since these decisions affect everyone. | Potentially, but limited. | Creation of environmental limits and targets may overtly identify cumulative anti-ecological bias in decision making. Limited ability for public input into proposed national planning frameworks. |
| Expand community involvement in major land-use and building decisions through public adversarial debates that can expose the long-term implications for public interests, including their redistributive outcomes and environmental impacts. | Potentially, but unclear. | This could be achieved at a local government/community level, but it is unclear how such debates would influence the addressing of environmental impacts. There is no overt requirement for this to occur. |
| Require that referendums and the similar events concerning development issues (sub-divisions, rezoning, etc.) or major new developments provide public fact sheets on pro and con positions that are agreed to by opponents and refer to further sources of information. | Possible, but unknown. | The NBA system enables this kind of approach, that is, public participation prior to finalising the plans or policies. However, this existed under the RMA, and the consensus positions had limited success addressing the cumulative environmental impacts, and any consideration of the life-cycle impacts were out of scope. |
| **Governance standards (s 7.2.3)** | | |
| Make sustainability and the maintenance of future options a fundamental human right since it affects every individual's and family's future, and make corporate and government sectors accountable for decisions irreversibly damaging the natural environment. | Potentially, but effects untested. | The component in the NBEB referring to the interconnectedness of all of the parts of the environment and the essential relationship between environmental health and its capacity to sustain life may favour this approach, but whether it makes a practical difference to decision making is unknown. |
| Convert development approval systems from rule-based processes and reductionist assessment tools that often concern only energy and resources (economics) into proactive frameworks that can address the ecological and ethical dimensions of sustainability. | Possible. | There is provision to engage in improving ecological integrity and moving beyond the bottom lines and the system's ability to absorb pollution. However, this is speculative, and rhe current wording does not require such an approach. |

**Table 1.** *Cont.*

| Positive Development Standard | Included in NBA System | Comment |
|---|---|---|
| Ensure that sustainability reporting, public information and community participation processes are sufficient to prevent environmental decisions being made 'informally' through subtle (yet, not illegal) forms of corporate and government collusion. | Unknown. | Not specifically addressed, but may evolve through the national planning framework. |
| **Urban Planning Issues at the Municipal or Regional Scale (s 8.2)** | | |
| Resource Security (RS) Analysis: Are the best locations for adaptable emergency facilities, environmental amenities and services identified to ensure universal security? | Potentially as part of spatial planning and a climate change response. | As part of the NBA system, these areas may be identified. |
| Risk Avoidance (RA) Analysis: Is the amount to be invested in preventative or corrective safety measures based on the worst-case scenario, rather than a gamble? Are there mapping opportunities for public benefits? | Potentially as part of the spatial planning and climate change response. | Some areas will be excluded from development and/or will require some form of managed retreat. The extent to which a worse-case scenario is applied is unclear. |
| Negative Space (NS) Analysis: Are the long-term impacts of the transfer of public space to private control (or vice versa) analysed and considered in urban policies? | Not addressed. | |
| Highest Ecological Use (HU) Analysis: Are the ecological deficiencies of the wider area that the site development could correct considered and addressed? | Potentially, but indirectly. | Work on environmental compensation may generate enhancement and restoration processes linking development with ecosystem integrity. However, this is speculative. |
| Ecological Transformation (ET) Analysis: Is the ecological evolution of regions from pre-urban to present times examined to identify appropriate species and ecosystems? | Potentially. | This could be incorporated as part of realising ecological integrity goals and applying limits, and may be achieved on a region-by-region basis, but there is no national requirement. |
| Ecological Space (ES) Analysis: Is there sufficient space set through a development's structures and/or landscapes for ecosystems and eco-services to offset the development's 'share' of ecological damage, which is caused directly and indirectly? | Unclear. | The NBA system includes provisions for integrated environmental management, provides for the protection of and the of appropriate restoration of the ecological integrity of natural systems, and this includes offset mechanisms. However, it is unclear the extent to which the impacts will be fully assessed, and the mitigation hierarchy and 'limits' approach to management does suggests that a full assessment will not occur. This is particularly the case when no mechanisms that currently exist require the assessment of the upstream effects (e.g., securing of materials, production of energy), and where the operational impacts (e.g., energy production) and downstream effects (e.g., decommissioning) have limited regularity oversight. |
| **Governance issues at the regional or national scale (s 8.4)** | | |
| Institutional Design (ID) Analysis: Do the performance indicators only reflect trends, or do they exclude comparisons relative to the remaining 'total' resource stocks and nature? | Potentially, but unclear. | Total environmental stocks will be taken into account, depending on how the NBA system is integrated and practiced. The extent to which different 'domains' are integrated in decision making is unclear. The "System outcomes" (s 5) are to be provided for in the national planning framework and all of the other plans, but what this might look like in pratice is unclear. |

**Table 1.** *Cont.*

| Positive Development Standard | Included in NBA System | Comment |
|---|---|---|
| Economic impact (EI) Analysis: Are the long-term costs of the ecological losses and resource depletion upon the economy itself (not just the financial costs) reported? | Not included within the scope of the NBA system. | |
| **Complexity and whole-systems impacts (s 10.7)** | | |
| Visualise the cumulative supply chain impacts, project lifecycles, and spill-over effects in different categories simultaneously. Include consideration of the embodied energy, water and carbon emissions during resource extraction and manufacturing. Include worsening conditions such as scarcity of land or depletion of resource stocks (groundwater, fertile soil, native forests, etc.) locally, nationally and internationally. | Downstream and upstream impacts of activities locally and nationally partially addressed. Impacts internationally not addressed. | The PD reference relates to the built environment, but it is relevant for all forms of development. The NBA system allows for direct development impacts on the ecosystems to be addressed, as well as the impacts on various domains, but it is unclear regarding the extent to which the upstream and downstream ecological implications are to be accounted for more broadly. |
| Supporting design for creating new symbiotic relationships to open up opportunities to increase ecological space. | Could be included within the NBA-SPA-CCA system, such as through the National Planning Framework integrating with spatial planning. | Relates specifically to the built environment in terms of designing characteristics that, if they are realised, could improve the functional habitat. This could also apply more generally in, e.g., emulating ecosystem functioning in the "working environments" (e.g., farms) to enable continuing food production while also improving other factors, e.g., biodiversity values and carbon sequestration and storage, and reducing emissions. This already happens at a voluntary/incentivised level in both the urban and rural systems in New Zealand. |
| Design should aim to increase human and environmental benefits in all aspects of a development and its surroundings. | Potentially enabled within the proposed NBA-SPA-CAA system. | Enabled to improve both human welfare and natural environmental values and ecosystem integrity. This may be compromised by the trading-off between welfare, values and integrity, and the application of the mitigation hierarchy. |

Table 1 suggests that the proposed NBA system will go some way toward creating a framework for increasing the awareness of the link between decision making and the need to increase ecological space by addressing ecological integrity. However, the means to deliver on this are limited or non-existent, and if they are left unaddressed, it is the overarching economic framework that will likely override the attempts to increase the ecological space.

Two contributors to creating the original *Resource Management Act*, Sir Geoffrey Palmer and Richard Clarke, advocate for a Natural Environment Act instead of an NBA to create a "framework that hangs over all of the various statutory regimes to connect them together with a common set of principles that are followed in all of the various contexts" [18] (p. 5). Palmer and Clarke's concerns focus on the range of the other acts that are not coordinated within the NBA system, as well as the fact that environmental protection will remain the responsibility of central and local government, whereas there needs to be more Parliamentary oversight. They also point out the risks of continuing economic pressures which ignore environmental costs.

Their concerns are partially addressed by requirements that regional spatial strategies to be created under the proposed NBA system are to have particular regard to relevant government policy statements generated through other legislation. The proposed SPA will also "promote the integration of the statutory functions associated with the management of the natural and built environments across multiple Acts" [19] (p.15).

The better integration of the statutory functions creates more rigorous ways to improve things under the current dominant paradigm framework, that is, making the current approach better. There will still be a reliance on establishing and enforcing limits, avoiding ecological harm, and using human well-being as a primary indicator of success. This is unlikely to generate overall net ecological benefits.

For example, Palmer and Clarke's proposed principles governing any new legislation include: promoting positive outcomes for the natural environment where they are practicable; identifying and avoiding, remedying or mitigating risks of ecosystem degradation or collapse whenever practicable; taking a precautionary approach where there is uncertainty coupled with potentially profoundly negative effects; carrying out environmental impact assessments to identify likely significant impacts; using economic instruments to ensure those causing damage avoid, remedy or mitigate the damage; requiring science-based solutions that address environmental issues; using relevant demographic information and policies. All of these principles currently exist, including some which are under the RMA, and they are unlikely to significantly alter the ecological outcomes.

Overall, the Palmer and Clarke approach continues the idea of responding to systems at the point of collapse, rather than requiring all development (not just harmful activities) contributes to improving ecological integrity and expanding biophysical functioning.

More helpfully, Palmer and Clarke point to non-Westminster-style jurisdictions such as those in Sweden, with its comprehensive and over-arching Environmental Code, and the numerous environmental or environmentally related international treaties and conventions which include principles and outcomes that are worth investigating. However, ultimately, there is a need for legislation to not just allow for improvements, but require them.

As highlighted by Palmer and Clarke [18], the *UK Environment Act 2021* arguably provides an evolution in thinking, and it is discussed in the next section.

## 6. UK Environment Act

The United Kingdom's *Environment Act 2021* provides targets for restoring natural systems. A particular example is the biodiversity gain objective in Schedule 14 where planning permission for a development in England provides for a condition that the projected biodiversity value attributable to the development exceeds by at least 10 per cent the biodiversity value of the onsite habitat prior to it being developed. The 10 percent is the total of the post-development biodiversity value of the onsite habitat, the biodiversity value of a registered offsite biodiversity gain allocated to the development and the biodiversity value of any biodiversity credits purchased for the development.

Underpinning this is work by the Department for Environment, Food and Rural Affairs (Defra) and Natural England to develop ways of measuring the biodiversity values using a habitat-based approach (the "biodiversity metric"). This enables planning permissions to continue while addressing biodiversity decline. However, there is limited empirical evidence that these approaches can deliver real gains, and the metric is still being tested, particularly in terms of its ability to contribute to coordinated landscape-level ecological gains [20].

This reinforces the ambiguous global data of offsetting efficacy, where the evidence is complicated by there being differing methodologies, difficulties proving additionality, and the promise of as-yet-to-be-demonstrated future gains for the current development permits [11,20]. This is balanced against the need to put in place a process that has the right credentials to fit within a neoliberal economy. In essence, the UK recognises that, as the development will continue, better accounting for the continuing biodiversity losses may deliver net gains. The positive view is that the investors and developers, in concert with non-governmental organisations, will recognise the need to not only comply with planning authority conditions, but also generate real biodiversity gains [12], responding to consumer demands and intergenerational changes in attitude.

While there is a depth of discussion to be had on the pros and cons of offsetting and net gain, the thrust of this article is focused on the place of the legislation in achieving the

PD outcomes. While there appears to be a genuine commitment within Natural England and Defra and the selected planning authorities, consultancies and developers to deliver real biodiversity improvements, these will remain marginal achievements in any country which prioritises the type of economic development that led to the ecological decline in the first place.

It should be noted that the New Zealand government is grappling with similar pressures, largely due to a significant infrastructure investment deficit [21]. While ecological integrity and human health targets will be mandatory, they will be set as with environmental limits "after taking into account other objectives, for example economic development, intergenerational equity and the risk of harm to ecosystems or human health" [21] (p. 9). This reinforces the likelihood that the targets will be traded off and net gain not necessarily required or achieved.

## 7. Proposed Ecological Integrity Legislation

The introduction to this paper claims that science may be used to inform the creation of policies, plans, legislation and regulation up until the information clashes unacceptably with the existing socio-economic system. From the discussion above, ecological integrity is an example of this. To recap, the NBEB defines ecological integrity as covering ecosystems, species, habitat and ecological functionality. NBEB s40 says that an environmental limit must be expressed as relating to the ecological integrity of the natural environment or to human health. The environmental limits are to be set as either a minimum biophysical state for an area or as the maximum amount of harm or stress, and in terms of ecological integrity, they must be set to reflect the current state of an area or the amount of harm or stress occurring in an area.

As discussed above, limits allow for but do not require improvements, they assume that the ecological components can be separately assessed, and they do not account for the ecosystem dynamics. They are static, whereas the science of ecology is premised on studying relationships and change. It is reasonable to observe that the law finds it difficult to articulate ways to manage change and uncertainty, but that is the law's problem.

This may be partly addressed in Section 5 of the NBEB which says the national planning framework and plans must provide for the protection or, if it is degraded, the restoration of the ecological integrity of a range of environmental domains or categories. The scale of the accumulated degradation in New Zealand [14] suggests few areas if any would have no degradation in at least one environmental domain. In addition, restoring ecological integrity implies the restoration of ecosystem processes, and the extent to which this is required depends on where the baseline is set.

The challenge in Section 5 is that integrity is set out as one of a number of "system outcomes" which includes providing for well-functioning urban and rural areas and various associated socio-economic outcomes. So, while in one sense it allows for addressing the degradation and restoration of these areas, this is also balanced against a number of other outcomes.

Given the above text, the following are some suggested changes, which recognise that that the law still needs to be structured and worded in a way that remains sensible to a liberal democracy in a global setting. If they are too radical, they will simply fail. The changes attempt to apply a more logical approach given that the premise is to address the cumulative losses and improve ecological integrity over time and space.

What follows focuses only on changes to the definitions and wording within the current proposed NBA. While these will not address all of the short falls identified in Table 1, they will increase the probability that these issues will be better addressed. The following suggested changes are modified from [11].

### 7.1. Concepts

The following should be included in the NBEB:

*Biophysical capacity*: There should be an explicit requirement for regulatory authorities to identify the extent to which biophysical capacity has been exceeded. This means that development would contribute to enhancing (meaning improving ecosystem functioning) and restoring (meaning recovery toward pre-development conditions) crucial life-supporting ecological complexes (in the same sense as used in [22]). This would need to be performed at a local level, while accounting for national or global implications. This does not address the full meaning of PD's ecological space, but it moves toward it.

The legislation needs to define and take account of the fundamental ecological principles of scale, interaction and complexity, the biogeochemical cycles and specificity of place, and the negative trends of disturbance, modification and fragmentation; contaminant accumulation and accumulated physical change; biodiversity decline. There are a range of indicators associated with these trends that exist or can be developed to identify the biophysical capacity locally, regionally, nationally and globally [23,24].

The capacity of a system is influenced by the extent to which the biophysical boundaries have been exceeded, or by contrast, where human activity is currently within the boundaries. One approach is downscaling the planetary boundaries analysis to suit New Zealand's conditions. Planetary boundaries are key biological and physical variables affecting the Earth's life support systems, and they include climate change, water, biodiversity and the flows of fertilisers such as nitrogen and phosphorus [25].

*Fragmented domains*: the draft proposes taking a fragmented approach to environmental domains (air, water, soils, coastal environments, wetlands, lakes, rivers, indigenous biodiversity and landscapes) despite collecting them within a section called System Outcomes (s5). While the NBEB enables an integrated management approach, it is unclear how this mix of ecological processes and components are to be managed as a system.

To make such integration manageable, the developers' contributions could span across domains and/or regions. For example, if a development impacts the environmental domains that are relatively healthy in a region, this may allow a contribution to assist the ecological restoration in other domains where the ecosystem values have declined. It would be necessary to outline the spatial and temporal extent of the transitory negative effects as part of this process, that is, the impacts to be avoided, remedied or mitigated during the development or initial operation. This would mean accepting that an investment in offsetting the impacts may not result in dividends until some point in the future [26] and accepting the risk that some initiatives may fail [12]. The framework allowing this to happen could occur through the proposed Spatial Planning Act and regional spatial strategies, particularly as they include 30-year timeframes.

While some regions may have significantly exceeded their local biophysical capacity, other regions, relatively speaking, may not. In some circumstances, this might mean modifying or excluding activities, or removing activities from one catchment or region to another. For example, the biophysical capacity of Auckland has been exceeded, measured by, for example, the decline in the health of the Hauraki Gulf which is caused in part by land-based activities [27]. Restoration may require the removal of some activities to different catchments or regions, or a process of de-intensification.

Again, this could be achieved through the proposed NBA system, but the enhancement and restoration goals should be required as a priority.

*Ecological net benefit:* where all of the development elements must demonstrate a proportional contribution to improving ecological processes. This means compensating for the negative impacts while also contributing to increasing ecological space. As noted, the transitional costs (ecological decline) over time and space will be allowed. It may be possible for the developers to invest in improving specific ecological processes before the development begins, though this would seem to apply only in a small number of cases. The *ecological net benefit* concept is similar to the biodiversity gain concept in the UK's *Environment Act 2021*.

*Ecological processes:* ecosystems create patterns that become apparent at a systems level, but they cannot be absolutely quantified at a component level. Investing in ecological processes (for example, through green infrastructure) generates appreciating assets which become better at delivering 'services'. It is accepted that under PD, investing in green infrastructure is not equivalent to increasing the ecological space. However, if they are designed well, these features can cumulatively increase benefits over time. The emphasis is on both design and maintenance in terms of success.

*Enhancement:* The means to facilitate the return of a species into an area, as well as supporting their co-existence and succession processes, by stabilising ecological functions through time [22]. This would directly contribute to increasing the ecological space, which is preferred under PD.

*Environment:* modify the current definition by removing reference to the natural environment and defining the environment as ecological processes and biotic and abiotic complexes.

*Restoration:* re-establish a species or habitat by direct action [22].

To help realise the above, the purpose of the act as it is currently drafted should change from enabling to *requiring* beneficial environmental outcomes and replacing the word "environment" with the more specific terms such as "*ecological processes*" and "*ecological net benefit*". The wording in italics is not in the current draft.

The current draft notes that people and communities should be allowed to use the environment in a way that supports the well-being of present generations without compromising the wellbeing of future generations. Instead, people and communities should be required to identify and work within the biophysical capacity of a district and region and account for planetary boundaries.

Achieving the purpose of the act should be through enabling the individuals and communities to protect, restore and enhance ecological processes, including as part of economic, social and cultural activities, and ecologically beneficial outcomes must be identified and pursued when they are required.

In addition, the mitigation hierarchy wording should be modified. Any adverse effects on the environment of its use must be avoided; where this is not possible, any activity must result in an ecological net benefit; in any case, all activities should contribute to an ecological net benefit. This is not clear in the current draft. A reference in the NBEB to the interconnectedness of all of the parts of the natural environment and the essential relationship between environmental health and its capacity to sustain life should be kept, replacing "environmental health" with "*functioning ecological processes*".

A reference to the limits as minima or maxima should be removed and replaced with a reference to improving biophysical capacity. Biophysical capacity is established by defining the extent to which regional activity is within the biophysical boundaries and where these boundaries are exceeded. This may include establishing quantifiable limits, but only as part of a system of continuing improvement. Where the boundaries have been exceeded, the activities contributing to those exceedances should be changed over time until the cumulative impacts operate within the boundaries.

### 7.2. Comment

None of the suggested changes are sufficient on their own to ensure an increase in ecological space, but they do provide a more effective framework, thus enabling this to be achieved. The real questions that arise relate to operationalising concepts such as biophysical capacity and ecological integrity.

A fair criticism is that the technical challenges attached to applying capacity and integrity concepts may undermine their usefulness. However, the premise that underpins applying such concepts in law is that they create a framework for a society to move beyond establishing limits and minimising harm toward creating conditions that will result in continual improvements in ecological capacity and functioning. As noted earlier, good

design and application should create the conditions for ecological assets that appreciate in value, obviating the need to quantify contributions relative to the impacts.

However, there are two remaining challenges, one being systemic, and the other relating to human capacity, as discussed in the next two sections.

## 8. A Systemic Problem

There is a need for aspirational goals to be underpinned by incentives as well as policed and enforced by rules. New Zealand has favoured incentives, particularly economic instruments, but not exclusively so. The tension between encouraging and requiring is relevant in terms of introducing legislation that is designed to generate improving ecological outcomes.

The first national policy statement (NPS) created under the 1991 RMA, the New Zealand Coastal Policy Statement, requires that local planning authorities ensure the Statement's objectives are adhered to. Equally, the NPS for Freshwater Management (NPSFM) and the associated environmental standards require compliance to protect and enhance water quality and manage its supply. There are minimum standards to adhere to, scope to improve ecological values, and clear relationships between land use and freshwater, wetland and coastal systems.

Unfortunately, environmental monitoring shows that there is continuing overall water quality decline in many areas [14], suggesting that such an environmental bottom line approach, even with scope for improvement, is insufficient.

New Zealand has also attempted an emissions trading system using the market allocation of greenhouse gas units. In this case, monitoring shows that the emissions keep increasing [28]. A Climate Change Response (Zero Carbon) Amendment Act in 2019 set targets and emissions budgets to improve the prospect of emissions reductions.

This mix of requirements and incentives has not led to achieving many ecosystem goals [14,27] in part because, as argued in this paper, there is a reluctance to put in place effective mechanisms. This comes down to having to justify actions within an economically rational framework [12], which while necessary in order to navigate through political reality, results in patchy progress toward achieving an overall ecological benefit.

Arguably, a symptom of this patchiness is the fragmented nature of national policy on the development and management of the natural environment. New Zealand's Parliamentary Commissioner for the Environment, commenting on the proposed NBA, illustrated this with the example of the mandating of medium-density housing clashing with efforts to realise better stormwater and infrastructure outcomes [29]. Legacy policy and actions that are being put in place under existing legislation will influence the development of infrastructures such as transport, housing and land use, thus creating an overhang which affects decision making even as the RMA is replaced.

The new legislation specifically addresses the need for long-term strategic and spatial planning and a systems approach. The first iteration of a national planning framework will include existing RMA national policy statements [19]. The objective is to create a more coherent and integrated system, identifying development options over decades.

However, as it was noted earlier, carried over into the new NBA system is the mitigation hierarchy and trade-off approach to development. Based on the existing trends and wording in the NBEB and SPB, a more integrated and strategic approach does not imply that there will be a net ecological benefit.

## 9. Who Will Do the Work?

Planning authorities and the private sector need the capability and commitment to design, implement, audit, monitor, assess and evaluate efforts to improve ecosystems. A UK survey undertaken in preparation for the passing of the UK Environment Act, particularly in relation to providing for the need to contribute to increasing biodiversity values, found that only five per cent of the responding planning authorities said they had sufficient resources to scrutinise all of the applications that might affect biodiversity. The

remainder reported they had no or very limited capacity to ensure most, if not all, of the applications were assessed by an ecologist. Very few planners had direct experience in assessing biodiversity net gain applications, and very few applications were sufficiently well prepared to be assessed in the first place [30].

Fewer than 10% of the respondents reported adequate expertise and resources to deliver net gain, while 85% of them reported a need for additional professional staff to support their new responsibilities. A quarter of them believe that they would only be able to address an increased net gain workload if other council activities that also require ecological input were reduced or additional resources were forthcoming. The majority of them reported that their capacity was inadequate to meet current needs, let alone additional ones [30].

The UK Government has committed to increasing the local government capacity [20], although the evidence to date suggests that this has yet to be realised (the UK survey [30] was commissioned by Defra and results released June 2022).

The situation in New Zealand is unclear, but there is likely not enough capacity to deliver on the increasing demands for assessing ecological integrity. Previous assessments of the New Zealand's capacity to deliver under existing compliance monitoring, evaluation, enforcement and reporting regimes indicate that there are significant shortfalls among some planning authorities due to in part a lack of expertise and staffing, but also due to poor political support and allocation of resources (see for example [31–35]). As of late 2022, the New Zealand Government was identifying the capacity and capability gaps [19], presumably with a view to addressing them.

## 10. Conclusions

This paper analyses the proposed changes to New Zealand's planning and environmental legislation, focusing on how this might increase the likelihood of a net ecological benefit. It uses the understanding and application of ecological integrity as an indicator of progress. The premise is that a scientific analysis is applied to develop policy and inform legislation up until the point that such logic is unable to be accommodated by the current dominant socio-economic paradigm.

To answer the three questions posed in the introduction:

(1)    Is the proposed legislative reform in New Zealand likely to deliver on the objectives stated in the legislation (the within the dominant paradigm (DP) logic frame)?

The Natural and Built Environment and Spatial Planning bills as tabled before Parliament will create a clearer framework for establishing the natural environmental limits within which development can occur. There will be an increased expectation to identify the medium- to long-term goals to improve certain natural environmental values, and this will be achieved at a local community level with central government support. Combined with planned climate adaptation legislation, this may generate more coordinated landscape-level ecosystem restoration. There is likely to be greater clarity over what the ecological trade-offs are when making economic and development decisions.

However, structurally, the planned legislation shows limited ability to deliver improved ecological outcomes, as defined by the draft legislation. The result is that while terms such as "ecological integrity" are referenced, it is unclear how the improvements in integrity will be realised.

(2)    Are the stated objectives anyway sufficient to deliver real net ecological benefit (the crossover from the within-DP to the within the Positive Development (PD) logic frames)?

The reasons for reviewing the legislation were that the current *Resource Management Act 1991* delivered neither good development outcomes nor good natural environment ones. The legislative review might improve the efficiency of delivering better outcomes, but it will not substantially alter the ecological integrity trajectory, which still follows the

mitigation hierarchy. That is, minimising harm and hopefully generating a net benefit where appropriate. In short, the answer to (2) is no.

(3)    What might the legislation look like when it takes a Positive Development approach (the within-PD frame)?

It is recommended that ecological integrity is the focus of any new legislation. This would assume that good design and implementation would see ecological integrity increase indefinitely. However, there would be no absolute quantifiable measure of success, and the outcomes, while they may be positive, would be uncertain. Meanwhile, increasing the biophysical capacity would be a more measurable metric that would act as a proxy for integrity.

The challenges include agreeing on what biophysical capacity means, and how it should measured, as well as the need to substantially increase the capacity of decision makers to assess, monitor and evaluate outcomes. The rationale is that progress in these areas will occur more rapidly if legislation is put in place requiring the understanding of the concepts as well as their application.

Overall, the legislation can contribute to achieving aspects of Positive Development, but it struggles to follow a line of reasoning that will result in real net positive ecological outcomes or increases in ecological space. The legislation should be treated as representing various transition states and providing greater or lesser support for communities wanting the shift toward a different framework, include the PD one.

**Funding:** This research received no external funding.

**Data Availability Statement:** Not applicable.

**Conflicts of Interest:** The author declares no conflict of interest.

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
