# Peer review of "Identifying Limits in Domestic Law Delivering Net Ecological Benefit: A New Zealand Example"

_urbansci, doi:10.3390/urbansci6040093_

Round 1

Reviewer 1 Report

The aticle deals with local conditions and this is the first remark. If the reader is interested in the subject of New Zeland legislation, this article is very interesting. However, I don't see any connection with contitions of a general nature. If we want to know the local conditions, tha analysis is detailed and interesting. 

The methodoloty and structure of the article is correct. 

I leave the publishing decision to the publisher. Article could be publish in such form or not published at all.

Author Response

Response: I have emphasised the international relevance at lines 76-83 and 118-123 in rewrite.

Reviewer 2 Report

Overall, the manuscript is very well written and argued. It is refreshingly candid about the extent to which well-intended legislation is likely to be reflected in good outcomes on the ground. The overarching point about the need for greater specificity/rigor in policy direction and, likely, resourcing is well made and rings true for me. Perhaps the author might allude to one or two other examples of this tendency in similar or other areas of resource management policy and legislation. That would help demonstrate that there are, in my mind, broader and more systemic issues with the way policy is often drafted. The overused word 'aspirational' comes to mind. 

The section entitled 'Proposed ecological integrity legislation' could be framed better, I think. It reads a bit like a submission on the legislation which is obviously not the function of the article. I think it could do with being re-framed a bit with more of an emphasis on the types of changes that could be made and perhaps how these might relate to other similar laws. I'd suggest some of the detailed tracked changes should be removed in favour of a more general analysis.

Might need to note that the exposure draft of the NBA was very skeletal. Therefore, it is hard to read too much into what the final legislation will look like. 

230-235 I didn't understand this paragraph. Re-write for clarity.

Given the international audience I would remove te reo names for concepts and just write in English instead, except where a direct quotation is necessary. Footnoted translations don't make for easy reading and add nothing to the analysis.

304 I might have missed it earlier, but this appeared as the first use of the concept of 'ecological space'. Could do with being defined here or earlier.

Author Response

I’ve added in a section (A Systematic Problem) at lines 671-712 discussing relevance to other policy and planning, including using the term aspiration. I’ve tried to strike a balance between sufficient detail while keeping things relevant for an international audience.

I have simplified suggested changes to the draft legislation at lines 620-653. Removed strike outs to improve clarity. The additional changes in the new A Systemic Problem section covers the need to comment on relevance to other legislation/policy. I’ve not looked at other NZ legislation in any detail as I don’t think that is necessary for the point of the article, and would be of limited interest internationally.

Re the skeletal nature of the legislation, see lines 513-521 and 747-753.

Review 230-235 re confusion. Rewritten now at 237-252 for clarity. This also cross-references to the new section A Systemic Problem noted above, as this helps provide further context for the commentary on the proposed legislation.

Regarding te reo: See lines 169-173 in rewrite plus section suggesting new wording at lines 620-652. Te reo references removed. Also removed reference to Te Ao in Table 1.

304 – first use of ‘ecological space’? Need to define here, or earlier – defined now line 89-90.

Reviewer 3 Report

First of all, I want to say that I enjoyed reading the paper, I consider it interesting and the subject is increasingly current. However, I would like to make some suggestions:

- At the moment, the economic guidelines of the government of Liz Truss are different from those written in the paper, so it is worth updating.

- There should be a greater review of the literature that, in addition to focusing on the UK Environment Act, focuses on that of countries other than Aglo-Saxon, because we are facing a global problem.

- Regarding the conclusions and regarding the second question of the paper, they are not clear enough. The conclusions should be more explicit with regard to the second question.

Author Response

Response: I have removed specific reference to Liz Truss at line 465-472 in rewrite.

Regarding a broader review of the literature, see an explanation for focusing on NZ at lines 76-83 and 119-124 in the rewrite. The point is to highlight how terminology introduced to seemingly address a problem may be confusing things if (a) definitions are poor and/or (b) undertakings to address the problem are confused, poorly linked, or lack legal weight. The paper is a case study to illustrate a broader point, and n international comparative planning review is not undertaken.

For clarifying the second question, see lines 784-794. This is largely left untouched as I do not feel it is ambiguous, but I have added a final line to be clear. Also I have formatted questions 1-3 to make it clearer to read.